# Evaluation of Acoustic Features after Refurbishment Works Inside Two Historical Opera Theatres Located in Italy

**Antonella Bevilacqua** [1] and **Lamberto Tronchin** [2,*]

1   Department of Architecture and Engineering, University of Parma, Area delle Scienze, 43100 Parma, Italy; antonella.bevilacqua@unipr.it
2   Department of Architecture, University of Bologna, Via Montalti 69, 47521 Cesena, Italy
*   Correspondence: lamberto.tronchin@unibo.it

**Abstract:** The acoustical characteristics of a room where the artistic performance is presented to the audience have a critical impact on the experience of both artists and spectators. It is important to know how the original aspects and the refurbishment works throughout the centuries are brought to the characterization of the sound field in such theatres, with positive and negative consequences. This paper presents the acoustical assessment of the Teatro Nuovo of Spoleto and the Teatro Alighieri of Ravenna, very important landmark centers for their historical and cultural activities. The acoustical characteristics have been gathered by placing the sound source on both the stage and orchestra pit, and the receivers in the stalls and balconies areas at different levels. It is of great interest to show the acoustical parameters of such Opera houses and some acoustical limits derived from intermediate interventions due to the need of the committees to allocate as many spectators as possible for income reasons. After an acoustic analysis of the existing conditions, the authors compare the acoustic behaviour inside the Teatro Nuovo of Spoleto using the image-source method (ISM) to investigate a change of ceiling configurations that occurred with the refurbishing works of the 20th century.

**Keywords:** acoustical measurements; Italian theatres; room acoustics; horseshoe box theatre





## 1. Introduction

The diffusion of pamphlets and treatises on the acoustical concepts during the 18th and 19th centuries are brought to the development of the modern theatre in Italy, signing the passage from a temporary to a stable architecture. This historical event involved many of the decisions taken from the committees, being composed of citizens rather than princes or royal families' leaders.

Sometimes, the choices taken across history in relation to the construction and refurbishment of theatres prevailed to the detriment of the good listening perception as designed by the first architects [1], determining important changes, sometimes considered unreversible, against the principles of the intangible cultural heritage [2,3].

This paper treats the historical background of two important modern theatres located in Italy [4–8]. The geometry and the architectural style have been delineated as well as the acoustical limits derived from non-technical committees during the modernization works [1,3]. A measurement campaign has been performed inside both theatres in order to show the objective acoustic parameters.

The measurements undertaken inside the Teatro Alighieri of Ravenna aim to highlight the importance of the resonance cavity beneath the pit floor and the relative beneficial effect for the acoustic parameters [9–13].

In line with standards and reviewing previous literature for performing measurements inside historical building [14–20], in both theatres an exponential sound sweep (ESS) has been used as a sound signal emitted by the loudspeaker to cover all the frequency spectrum.

For the Teatro Nuovo of Spoleto, further analysis and acoustic comparison between the existing and original ceiling configurations have been introduced [21–30]. In particular,

the image-source method (ISM) has been applied in combination with the analysis of EDT and ITDG parameters [31–34], methods employed for understanding the lateral and vertical reflections at the stalls and balconies areas [35–43].

## 2. Documentary Background

### 2.1. The History of the Teatro Nuovo of Spoleto

The Teatro Nuovo of Spoleto was opened in 1864, based on the project led by the architect Ireneo Aleandri [4]. Until the 18th century, the city was provided with a theatre considered too narrow to accommodate an always growing audience interested in artistical shows [5]. As such, the necessity to build a new larger theatre, to be placed in the town center, was supported by the position of 96 citizens who contributed with their savings to cover the costs of the construction materials as a primary need to start to edificate [6].

The local authority decided that the Teatro Nuovo could be erected in place of the monastery of St Andrew and the near church, which occupied most of the space that should be instead dedicated to the open square [6].

The orchestra pit was realized in 1914 when the floor of the original stage had been cut off to allocate the musicians on a lower level than the audience area [7]. Further modifications were applied to the orchestra pit during 1950, aiming to extend the space for the orchestra beneath the stage floor. Figure 1 shows the internal views of the balconies and the stage.

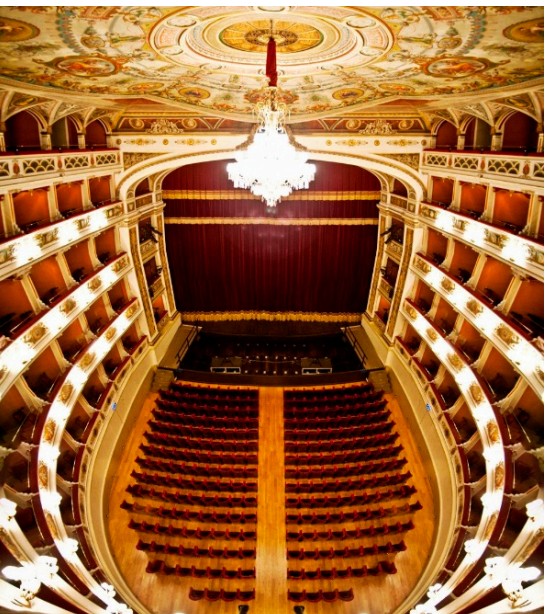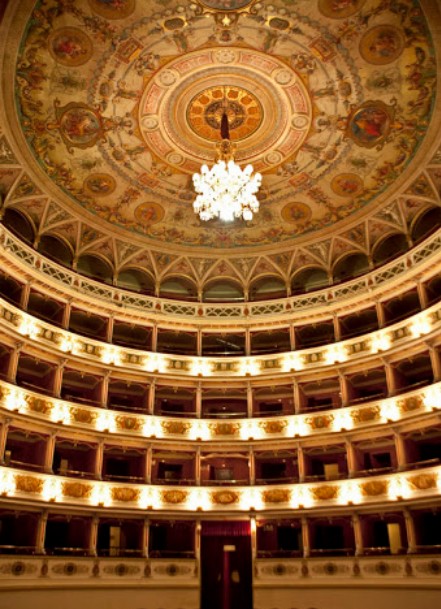

**Figure 1.** Perspectival views of the Teatro Nuovo of Spoleto.

Nowadays, the theatre is considered the referent artistical center of different national festivals. The most important event is the Festival of the Two Worlds, created by the conductor Gian Carlo Menotti (1911–2007), who donated the statues of four literature men (i.e., Rossini, Alfieri, Goldoni, and Metastasio) which are still located in the niches of the arched porch of the entrance [7]. For this gesture, other than his great contribution to the development of the musical activity of Spoleto, the theatre was dedicated to G. C. Menotti, based on a city council decree in 2010.

Between 2003 and 2006, taking advantage of a general renewal of the safety conditions inside the theatre, a long campaign of restoration works took place, to comply with the new standards and regulations. The works included the structural consolidation of the balconies, the provision of fire exits, the placement of a new electrical system, as well as the restoration of wooden beams and the ceiling of the main hall. Additionally, the

introduction of machinery to facilitate activities on the stage occurred, as well. Based on the agreement proposed by the city council, the contractor was requested to conduct the works preserving the original architectural characteristics, avoiding any alterations to the finished materials and what could affect the acoustics [6].

In a previous time period, specifically in 1933, some modifications included the substitution of the wooden beams supporting the stage with a steel frame structure, as it can be found in many other Italian Opera houses [7].

### 2.2. The History of the Teatro Alighieri of Ravenna

The Teatro Alighieri of Ravenna has a similar historical background. In fact, in 1838 the city council deliberated the construction of a new theatre in place of the old one named *Comunicativo* [8]. As such, the project was assigned to Tomaso and Giovan Battista Meduna, the architects that were in charge of refurbishing the theatre La Fenice of Venice. The two architects were supported by the painters G. Voltan and G. L. Gatteri for the artistic decorations made in neoclassic style [8]. The theatre had flourishing seasons for both dramas and Operas, running continuously until the First World War, when it had been closed for the first time [8].

After the First World War, the artistical performance ran until 1929, when the theatre was closed for refurbishing works that involved the consolidation of the wooden frame of the balconies and the widening of the stage. Furthermore, the stalls and the stage were completely remodeled, while the seat upholstery was refitted and the electrical system was incorporated in the existing construction [8]. Figure 2 shows some views of the internal organization. In 1967, four statues were placed in the niches of the entrance: they represent the symbols of four arts (i.e., Tragedy, Melody, Dance, and Music).

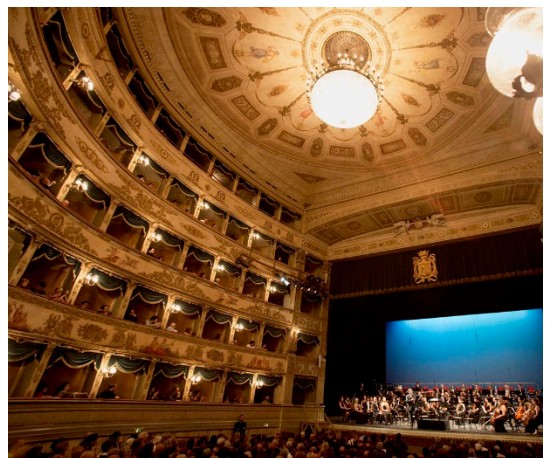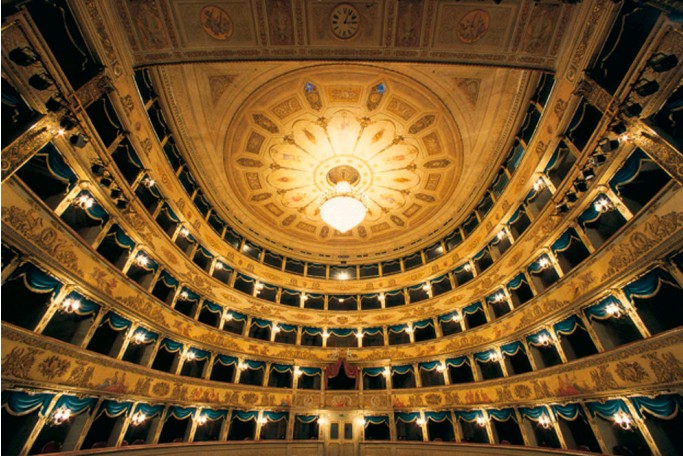

**Figure 2.** Views of the Teatro Alighieri of Ravenna.

Given the common practice during the 18th century to provide the machinery to lift the pit floor at different levels, the attempt to fill and substitute the void with mechanical systems failed. Therefore, the Teatro Alighieri represents one of the few theatres in the world still preserving the resonance cavity beneath the pit floor [9]. The cavity has the shape of a wooden hull with the ribs protruding outwards, representing a construction element usually realized by the Venetian carpenters. The resonance cavity beneath the pit floor is found to be structurally similar to the cavity existing below the stalls floor [9].

During the 90s, the Teatro Alighieri was subject to further renovation and adaptation works to comply with the seismic and fire requirements. A further modernization work undertaken in the new millennium consisted of the installation of a removable acoustic chamber. This latest light structure is composed of three fixed sides and closed on top by a ceiling equipped with wooden panels, useful for increasing sound diffusion [10]. The

two lateral panels of the acoustic chamber have a slight vertical inclination in order to accentuate the stage perspective.

On the same occasion, a hydraulic machine was installed in the backstage area for facilitating the technical and practical activities of lifting the scenery [9].

## 3. Architectural Organization

### 3.1. Geometry and Architectural Features Inside the Teatro Nuovo of Spoleto

The two theatres discussed here are very similar in shape and volume size.

In particular, the Teatro Nuovo of Spoleto is considered the most important of its region, having a capacity of 280 seats in the stalls and 320 seats on the elevated boxes [6,11]. The stalls are separated by a corridor running along the longitudinal axis and are coronated by four orders of balconies, surmounted by a gallery having a capacity of further 200 seats occasionally occupied. The boxes were designed larger at the center and narrower towards the proscenium [6].

The ceiling of the main hall is composed of a light structure made of curved canes, anchored to the wooden beams frame. The junction of the ceiling to the gallery has the shape of an umbrella, evoking the style of the architect Vanvitelli, criticized at the time of its realization [11].

The stage has dimensions of 29 × 20.5 m [L × W] and the proscenium arch is 12 m large. The architect Aleandri, who also designed the Sferisterio of Macerata, drew the main hall of Teatro Nuovo with a plan layout having a horseshoe shape, as visible in Figure 3 [11].

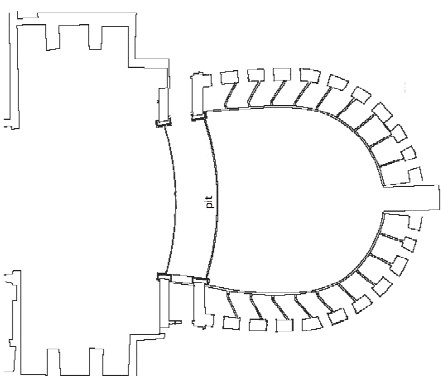

**Figure 3.** Plan layout of the Teatro Nuovo of Spoleto.

Figure 3 shows the architectural organization of the Teatro Nuovo, where the floor of the stalls is slightly inclined.

### 3.2. Geometry and Architectural Features Inside the Teatro Alighieri of Ravenna

The initial design of the Teatro Alighieri shows an elliptical hall that has been transformed successively to a horseshoe shape after the refurbishing works (see Figure 4). The theatre is composed of four orders of 25 balconies, with a gallery at the top-level crowning the perimeter of the main hall [9]. After the works of the 20th century, the stalls were allocated on a slightly inclined floor, with a capacity of 316 seats against the 520 distributed onto the balconies. The gallery on the top level can host up to 190 spectators, but it is rarely filled.

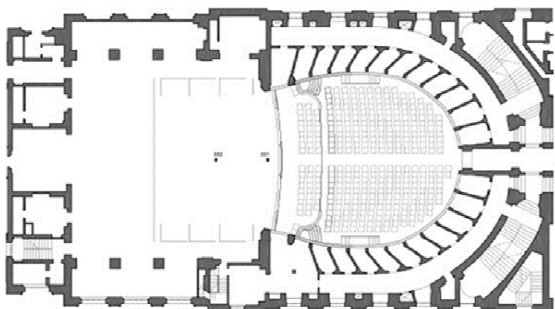

**Figure 4.** Plan layout of the Teatro Alighieri of Ravenna.

A continuous line between the three rows of seats and the rest of the stalls is visible in Figure 4. This indicates a different function of the seats closer to the stage. In particular, the pit floor can be lifted at the same level of the stage when more capacity is required, especially during the performance of symphonic orchestras. Then, it can stay at the same level of the stalls' floor or even below, when it is required for Opera. This ability to choose different options has been introduced in order to accomplish all the provisional needs.

### 3.3. Functional Differences and Analogies

The Teatro Nuovo of Spoleto in terms of audience capacity is slightly smaller than the Teatro Alighieri. From a geometrical point of view, the stage in Spoleto is wider transversally, whereas in Ravenna it is almost squared. The plan layout of the main hall has a shape of a horseshoe box in both theatres.

Table 1 below summarizes the different features between the two theatres.

**Table 1.** Architectural characteristics of the Teatro Nuovo and the Teatro Alighieri.

| Description | Teatro Nuovo (Spoleto) | Teatro Alighieri (Ravenna) |
|---|---|---|
| Type of plan layout | Horseshoe box | Horseshoe box |
| Total capacity (no. of seats) | 800 | 1026 |
| Inclination of stalls area (%) | 5% | 5% |
| Stage dimension (m) [L × W] | 29 × 20.5 | 25 × 22 |
| Volume (m$^3$) | 6300 | 7100 |

In the history of a performing arts place, the restoration works was not always able to deal with the indoor optimum acoustics. This is explained in the following subsections.

#### 3.3.1. Alterations Applied in Teatro Nuovo

In Spoleto, the addition of the gallery on top of the balconies forced the roof ceiling to have an accentuated curvature (rise of 1.3 m at the center), provoking the sound reflections to be changed in directionality [11]. Additionally, the retreat of the stage floor to widen the orchestra pit excluded the singers' voice to be well reinforced by the construction elements of the proscenium [11]. The pit itself has been extended for 13 m beneath the stage floor [6,11], a space so deep that the reflections from the back wall are too delayed and might be perceived as echoed by the musicians situated in the pit [6,11]. Further considerations are given in Section 6.

#### 3.3.2. Potential Changes Tested in Teatro Alighieri

Fortunately, in Ravenna the intention to fill the resonance box below the pit had been neglected since acoustic experiments undertaken in 1992 and 1994 by researchers from the University of Bologna demonstrated the worsening conditions of the overall acoustics by filling the resonance box [9].

In particular, the first experiments of 1992 consisted of recording four musical performances executed by a group of strings and wind instruments players as well as a pianist.

The same performances were played with the cavity empty and filled with wood saw-dust. The subjective results were issued by a pool of professionals in music, digging the preference on the empty cavity [9].

In addition to the judgments of the musicians, acoustic studies of the Teatro Alighieri were undertaken by the same researchers 2 years later (1994) in order to evaluate the importance of the resonance box located beneath the pit floor [9]. The measurements were undertaken with the cavity empty and successively filled with compressed polystyrene mats. The equipment used during this survey is listed as follows:

- Omnidirectional sound source (B&K4224);
- Binaural dummy head (Sennheiser MKE2002 set);
- Radiomicrophones (Nady 201 VHF—LT Lavalier Bodypack Transmitter, Wireless System Receiver);
- MLSSA audio board (A2D160) and related software (MLSSA 10.0C) for the post-processing;
- Personal Computer (Toshiba 3200).

Comparing the acoustic parameters obtained from the impulse responses (IRs) taken during the two conditions (i.e., with the cavity filled and empty of any material), the following outcomes have been obtained:

- The sound levels across the theatre with the empty cavity were increased by 5–8 dB [9,12];
- The values of EDT, $T_{15}$, and $T_{30}$ were constantly increased by 15–20% at each frequency band with the condition of the empty cavity [9];
- The clarity index for music ($C_{80}$) resulted as being invariant, with a negligible drift between the two conditions [9].

Overall, considering the average values of all the measured points, as shown in Figure 5, the results indicate that the sound is more robust with the empty cavity, especially regarding both the reverberation time and the sound pressure levels [12]. These outcomes are in line with the subjective response perceived by musicians in 1992 [13].

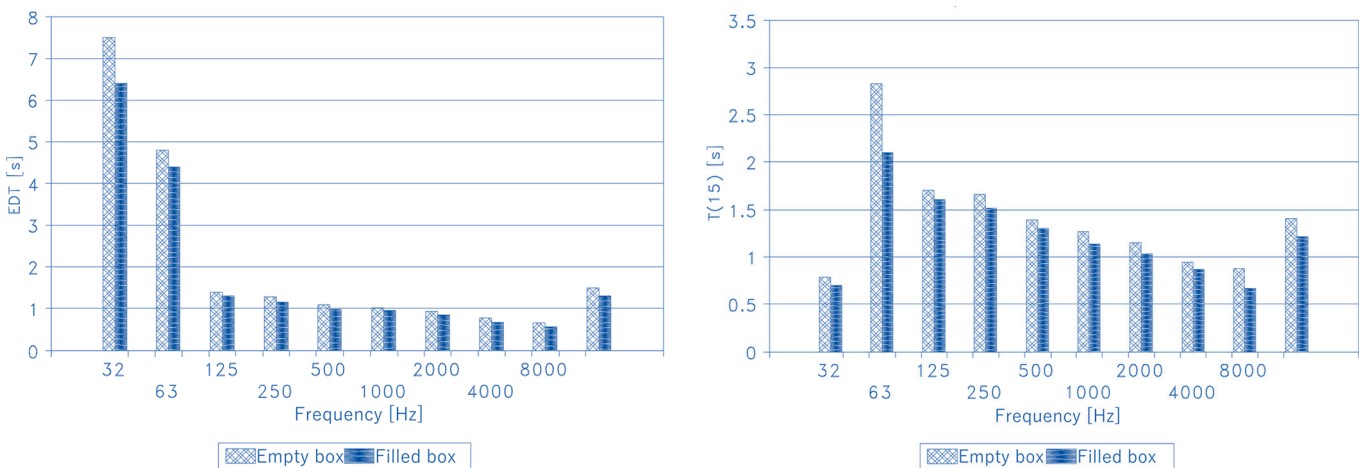

**Figure 5.** *Cont.*

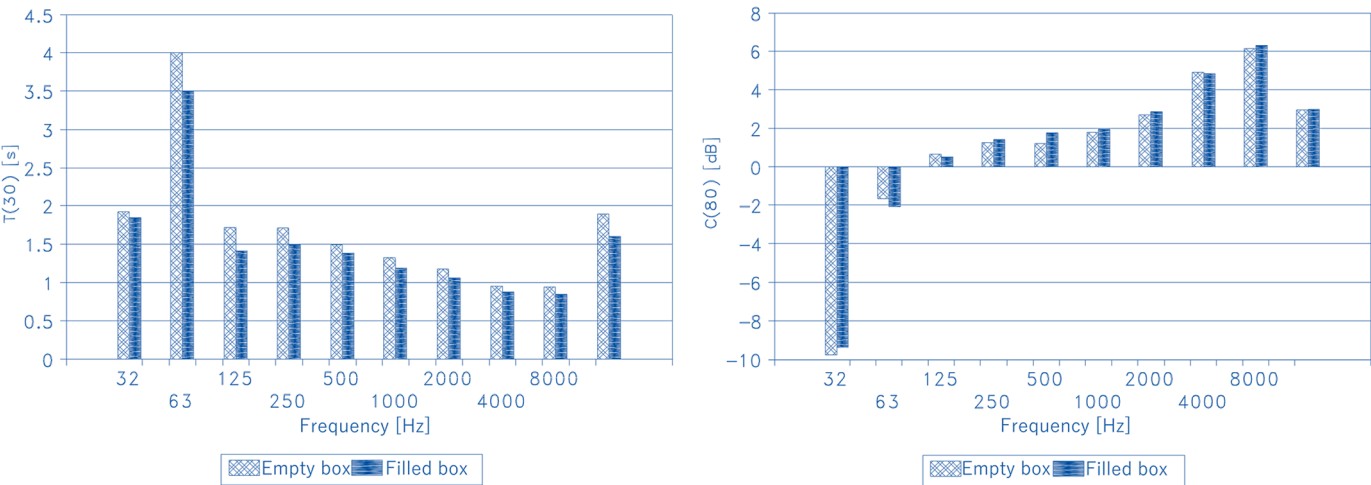

**Figure 5.** Acoustical parameters inside the Teatro Alighieri of Ravenna. Comparison between the empty and filled resonance cavity beneath the orchestra pit.

## 4. Measurements

Acoustic surveys were undertaken inside both theatres to understand the acoustic behaviour of the existing volume and material conditions after historical alterations. Objective parameters were obtained in line with the standard recommendations outlined in ISO 3382-1 [14]. During the surveys, therm-hygrometric conditions were taken in consideration [15]. The sound source was placed at 1.4 m from the finished floor, while the receivers where installed at the height of 1.2 m on stalls and boxes. In both theatres, the measurements were undertaken in unoccupied conditions and without any scenery mounted.

### 4.1. Equipment and Instrumentation

The equipment used in both theatres includes the following [14]:

- Pre-equalised omnidirectional loudspeaker (Look Line);
- Binaural dummy head (Neumann KU-100);
- B-Format microphone (Soundfield MK-V);
- Personal Computer connected to the loudspeaker and the microphones.

The excitation signal to measure the RIR was an exponential sine sweep (ESS) having a frequency range between 40 and 20 kHz [16,17].

### 4.2. Sound Source and Microphones Positions in Spoleto

In the Teatro Nuovo of Spoleto, the acoustical measurements were undertaken by placing the sound source on both the stage and orchestra pit, while the receivers have been placed in 38 positions in the stalls and inside 14 selected balconies, as shown in Figures 6 and 7.

In the theatre of Spoleto, given the symmetry of the volume, the measurements in the balconies have been undertaken only on one side of the main longitudinal axis. The selected receiver positions represent all of the audience area [18].

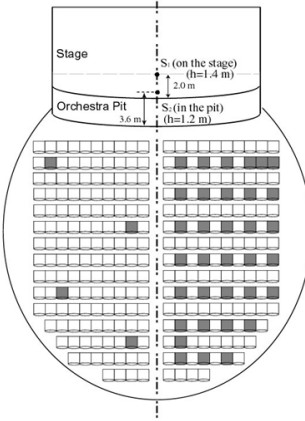

**Figure 6.** Scheme of the equipment location during the acoustic measurements across the stalls of the Teatro Nuovo of Spoleto. The grey-colored squares indicate the receivers positions, while the blue dot and pentagon indicate the loudspeaker positions, respectively on the stage and in the orchestra pit.

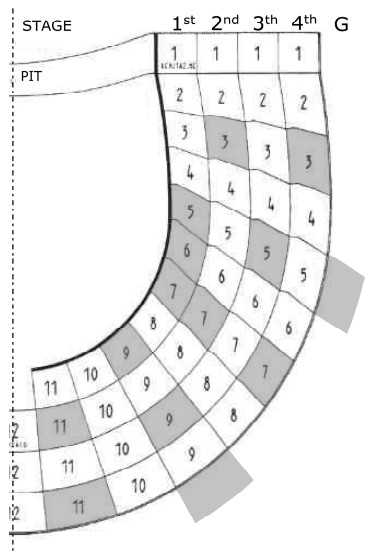

**Figure 7.** Scheme of the equipment location during the acoustic measurements across the levels of boxes inside the Teatro Nuovo of Spoleto. The grey-colored squares indicate the receivers positions, while the cardinal numbers represent the box numbers at each order (i.e., first, second, third, fourth), as indicated on top, including the gallery (G).

*4.3. Sound Source and Microphones Positions in Ravenna*

In the Teatro Alighieri of Ravenna, the acoustical measurements were undertaken by placing the sound source on the stage, at the center of the proscenium [19], while the receivers have been placed in 44 locations across the stalls and inside 12 selected balconies. Figures 8 and 9 show the source and the receivers' positions during the acoustical measurements.

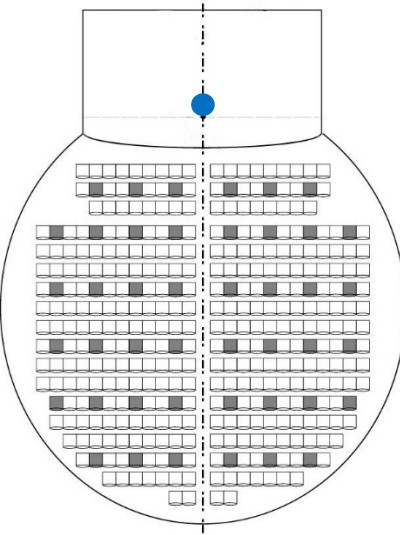

**Figure 8.** Scheme of the equipment location during the acoustic measurements across the stalls of the Teatro Alighieri of Ravenna. The grey-colored squares indicate the receivers positions, while the blue dot indicates the loudspeaker positions on the stage.

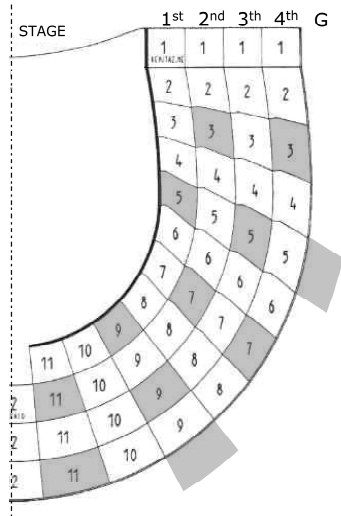

**Figure 9.** Scheme of the equipment location during the acoustic measurements across the levels of boxes inside the Teatro Alighieri of Ravenna. The grey-colored squares indicate the receivers positions, while the cardinal numbers represent the box numbers at each order (i.e., first, second, third, fourth), as indicated on top, including the gallery (G).

In Ravenna, since the plan layout is specular with respect to the medium axis, the selected receiver positions would be characterizing all of the sitting areas [18].

## 5. Measured Results

The graphs in Figure 10 are obtained by considering the average values of all the measurements' positions.

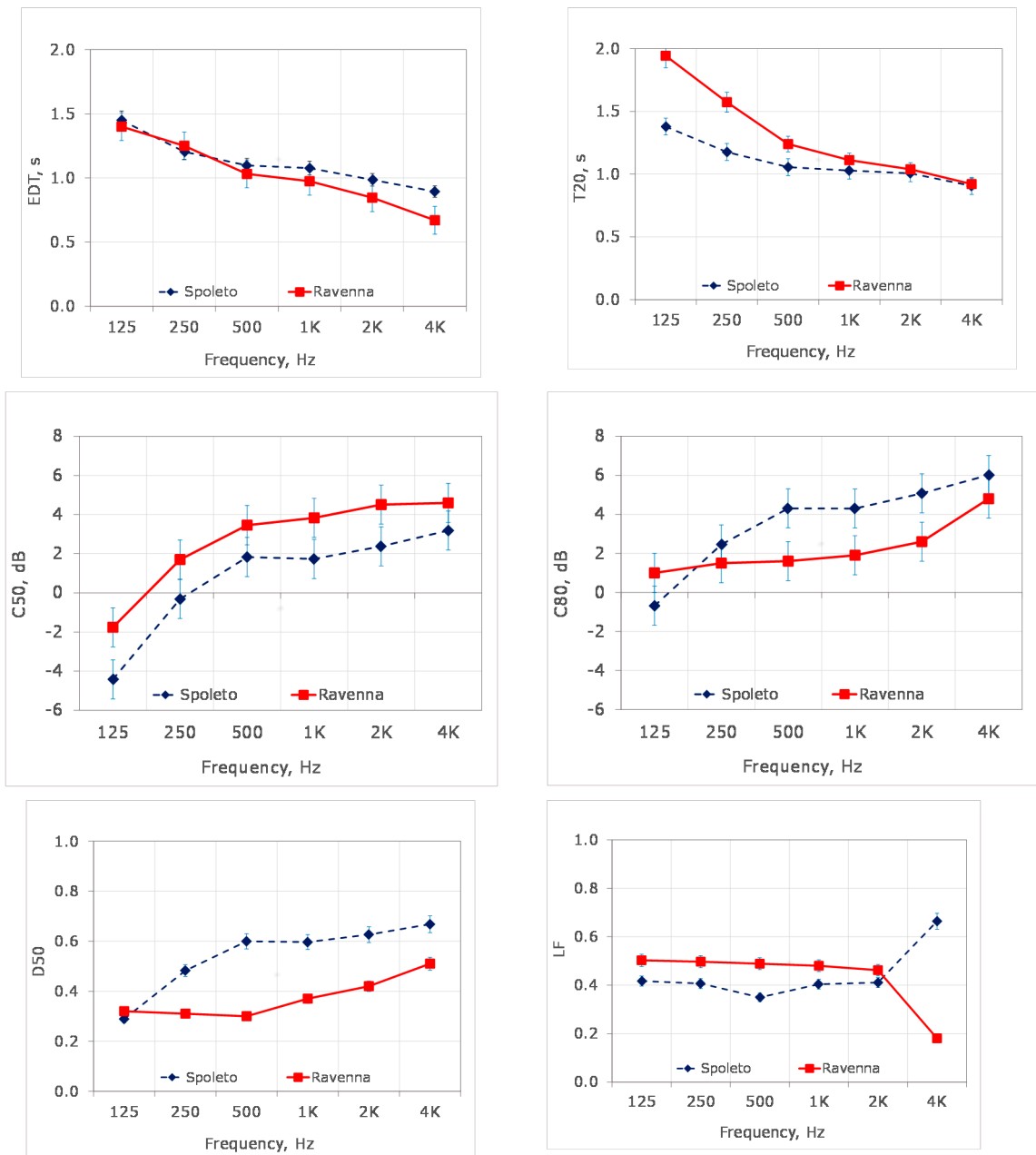

**Figure 10.** Comparison of the main acoustical parameters between Teatro Nuovo of Spoleto and Teatro Alighieri of Ravenna.

### 5.1. Early Decay Time (EDT)

The graph related to EDT shows that the trend line of the two theatres is very similar to each other. For mid-high frequencies, the values of Spoleto are 0.2 s higher than those measured in Ravenna, while for 125 Hz the difference is less than 0.1 s, within the just noticeable difference (JND) limits. The optimal values of EDT, as defined by Jordan [20], range between 1.8 and 2.6 s, which are not achieved in both theatres. The sound perception inside such environments, in terms of a subjective evaluation, could be defined as slightly "dry" or "deaf" [21]. Furthermore, the increase in reverberation time at a low frequency is explained by the absence of audience, which would be absorbing much more reflections of sound waves than empty seats, contributing to the reduction and balance of the booming effect [22,23].

### 5.2. Reverberation Time

Based on room volume and reverberation time, it is possible to see that the parameter $T_{20}$ in both theatres results slightly below the target of optimum values [24], as indicated in Figure 11, even if the overall values result within the JND established for reverberation time [25]. These results are due to the decisions taken by the committees in agreement with the architects during the phase of design [6–8], with the intention to make the theatres suitable for both music and prose performance [6–8]. In addition, the refurbishment works occurring during 1933 inside the Teatro Nuovo and 1929 inside the Teatro Alighieri, regarding the extension of the orchestra pit in Ravenna and the finished materials in both theatres, affected the volume and the averaged absorption coefficient, hence the $T_{20}$ [26].

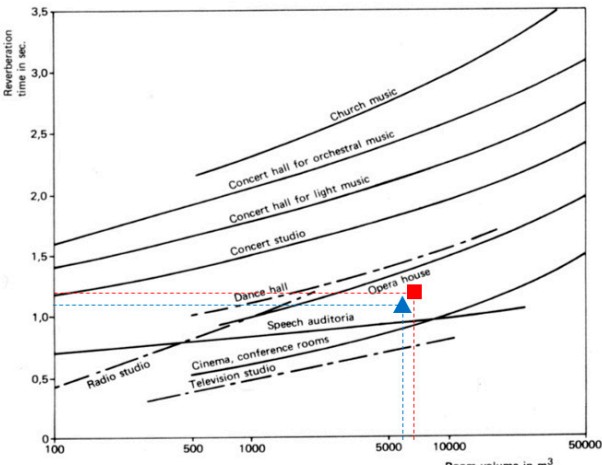

**Figure 11.** Optimum reverberation time values in the function of the room volume [27]. The blue triangle is related to the Teatro Nuovo of Spoleto and the red square to the Teatro Alighieri of Ravenna.

The line trends of $T_{20}$ in Figure 10 indicate a maximum difference of 0.5 s at 125 Hz between the two theatres, which become smaller towards the high frequencies.

### 5.3. Clarity Indexes ($C_{50}$ and $C_{80}$)

The trend line of the speech clarity index ($C_{50}$) in Ravenna is higher than in Spoleto, with values at mid-high frequencies meeting the target of >3 dB, as defined by the literature [28]. Given the results in the Teatro Nuovo, the value of $C_{50}$ at 125 Hz are 5 dB below the lowest JND limit, and in Teatro Alighieri the value of $C_{50}$ at 125 Hz is 4 dB below the lowest JND limit. This might indicate that there could be light difficulties in speech understanding at this frequency range.

In terms of music ($C_{80}$), the clarity index results are moderately higher in Spoleto, particularly at mid-high frequencies. The values of $C_{80}$ should not widely overcome the upper range limit (i.e., 2 dB) [24,28]: In Ravenna, the value of $C_{80}$ at 4 kHz exceeds 1.5 dB which is the upper range limit of 2 dB, while in Spoleto the $C_{80}$ exceeds the upper range limit of 2 dB from 500 Hz onwards, thus exceeding about 2 dB at 4 kHz. In both theatres, the values at low frequencies are within the optimum target [28].

### 5.4. Definition ($D_{50}$)

A good speech definition is achieved for values higher than 0.5 (i.e., 50%), while the optimum values for music definition are lower than 0.5 (i.e., 50%) [29]. On this basis, the results indicated in Figure 10 show that the values of $D_{50}$ in the Teatro Alighieri are good for music definition, while in the Teatro Nuovo all values of $D_{50}$ are in the optimum range for speech, except at 125 Hz, where the result of 0.3 (i.e., 30%) is within the optimum range of music definition. In other words, at low frequencies the $D_{50}$ results are more appropriate

for music performance in both theatres, while at mid-high frequencies the $D_{50}$ results are suitable for both speech and music, particularly in the Teatro Nuovo [29].

### 5.5. Lateral Energy Fraction (LF)

The LF is an important acoustic parameter that correlates the sense of spaciousness of a listening space to the music for those in the audience [30]. The lateral reflections in a performance arts space are important to give the listeners a sense of immersion and the construction elements that contribute to providing this effect are the lateral walls of the proscenium and, depending on geometry and type of volume, the vertical walls of the balconies [31]. A good value in the literature [31] is considered LF > 0.25 (25%). In both Spoleto and Ravenna, the LF results are in the optimum range across all the frequency bands, except at 4 kHz for the Teatro Alighieri.

### 5.6. Acoustic Maps

We can also see the results considering a different way of representation, which indicates the spatial distribution of the acoustic parameters across the site. The graphs in Figure 10 give a global view of the acoustics inside the theatres, while the maps show how the acoustic parameters vary across the main hall.

The acoustic maps have been realized using the software MATLAB 2019, which interpolates the measured values in combination with the plan layout of the space. The grid metric has been set to 5 m, with the coordinates (0;0) corresponding to the center of the stage on the medium axis. Each map represents the plan view at 2 m high, where the color change of the scale is marked by isometric lines.

Figures 12–14 report a few acoustic parameters with the spatial distribution of all the averaged measured values at 500 Hz, a frequency band suitable for evaluating both speech and music.

In particular, Figure 12 shows that the center of the main hall of both theatres, designated by the yellow area, is the most reverberant zone. The same part, as shown in Figure 13 in a grey-shade color, corresponds to the best clarity index zone, with $C_{80}$ equivalent to 0 dB.

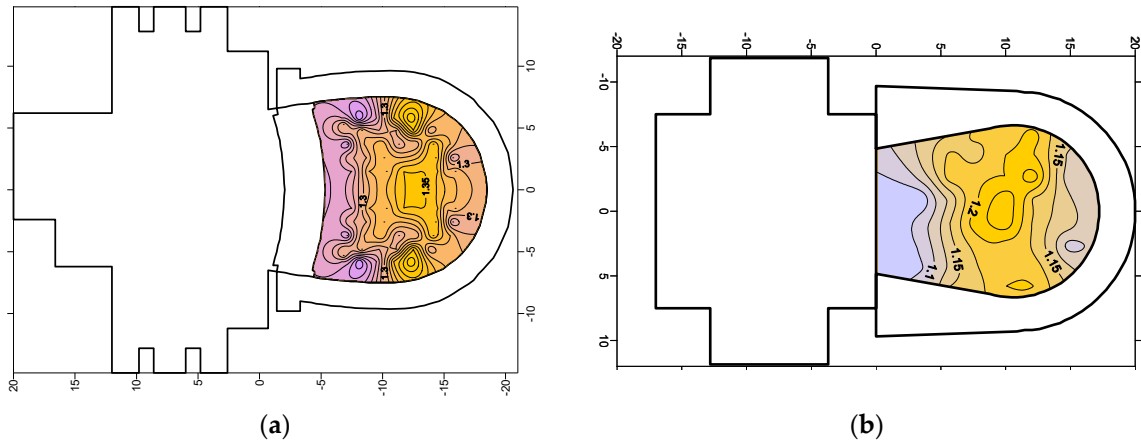

(**a**)        (**b**)

**Figure 12.** Spatial distribution of $T_{20}$ at 500 Hz: In (**a**) Teatro Nuovo of Spoleto and (**b**) Teatro Alighieri of Ravenna. The color scale represents the variation of the range between 0 s (grey-pink) and 2 s (yellow). The contour lines are stepped of 0.02 s.

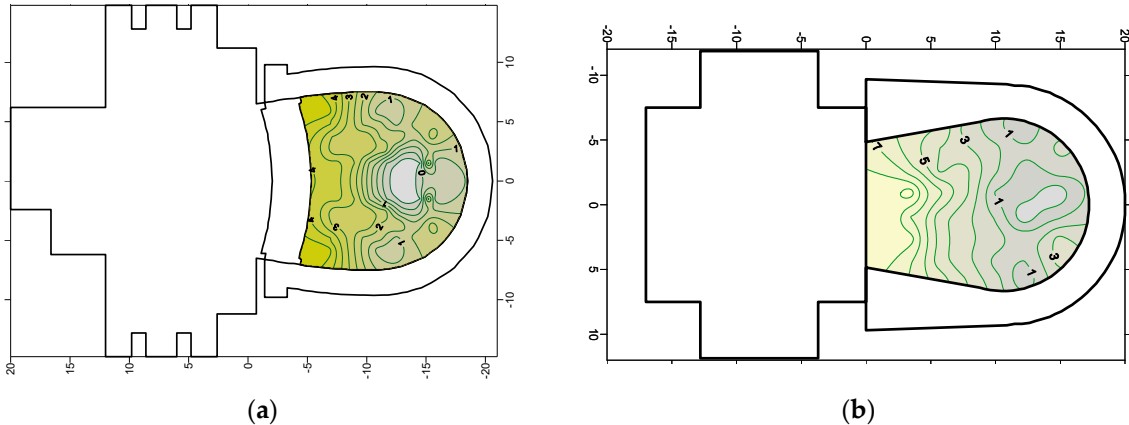

**Figure 13.** Spatial distribution of $C_{80}$ at 500 Hz: In (**a**) Teatro Nuovo of Spoleto and (**b**) Teatro Alighieri of Ravenna. The color scale represents the dB variation between −1 (grey) and +7 (yellow-green). The contour lines are stepped of 1 dB.

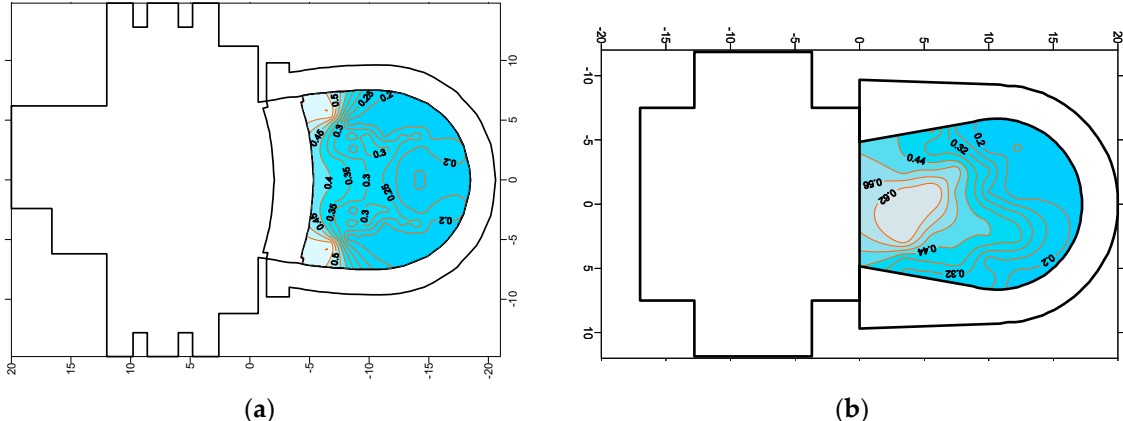

**Figure 14.** Spatial distribution of inter aural cross correlation (IACC) at 500 Hz: In (**a**) Teatro Nuovo of Spoleto and (**b**) Teatro Alighieri of Ravenna. The color scale represents the variation between 0 (blue) and 1 (white).

A complete explanation of the spatial distribution, related to the sound energy across space, is herein introduced by the inter-aural cross-correlation (IACC). The IACC is defined in the literature [32] as the measure of the diffuseness of the sound field in a room, contributing to assess the envelopment of sound at the listener position. The lower the value of IACC, the better the sense of envelopment. Figure 14a,b shows that the best value of IACC is found at the last rows of seats far from the stage, to be specifically 0.2 in both theatres. As long as the listener moves towards the stage, the IACC becomes lower, meaning that the directionality of sound rays is not horizontal or lateral, but vertically due to the reflections coming from the floor or the ceiling, worsening the sense of immersion in the reverberant field [33].

## 6. Acoustical Limits in Teatro Nuovo of Spoleto

In Teatro Nuovo of Spoleto, a second campaign of measurements was undertaken in order to study the reflections inside the theatre by placing the sound source in 20 positions on the stage and in 20 positions in the orchestra pit, while the microphone was placed in the stalls and on the third order of balcony, as shown in Figure 15.

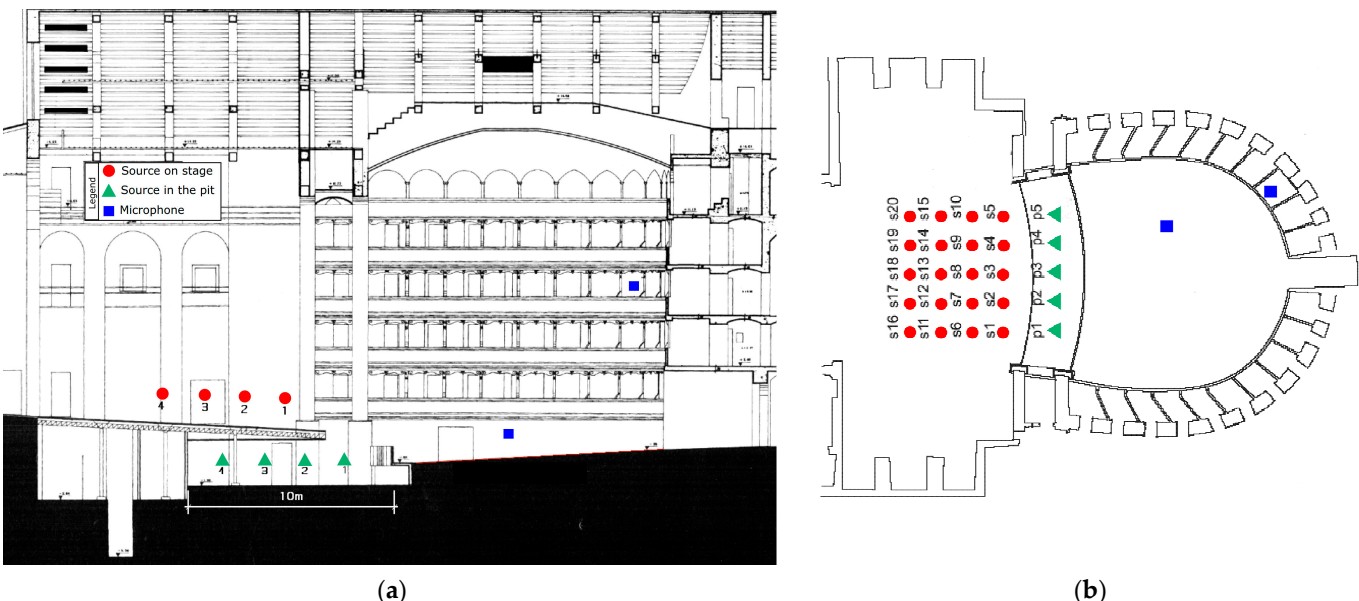

**Figure 15.** Equipment positions in the Teatro Nuovo of Spoleto during the second campaign of measurements. Red dots represent the sound source on the stage; green triangles represent the sound source in the orchestra pit; blue squares represent the microphone positions, as shown in section (**a**) and in plan (**b**).

The sound source was at the height of 1.4 m from the stage floor and 1.2 m high from the pit floor, while the microphone was standing at 1.2 m from the relative floor. The equipment selection is the same as described in Section 4.1, and the excitation signal was the ESS having a frequency range between 40 and 20 kHz [16,17].

For the study on the reflections, the EDT and the initial time delay gap (ITDG) parameters have been selected for the following reasons:

- The EDT represents the decay time of the first 10 dB of the sound energy emitted in the room by the impulse, and consequently it is very sensible to the geometric details of the room volume [34];
- The EDT has been used by Gade in many binaural RIR measurements for performing arts spaces [35];
- Since the EDT becomes lower by increasing the distance between the source and the receiver, it is considered the appropriate parameter to compare different positions in the same room [34];
- The ITDG is defined as the time between the arrival of the direct sound and the early reflections of maximum amplitude [35]. It describes the overall characteristics of a hall, allowing also a potential recognition of the reflecting surfaces contributing to the early reflections [36].

*6.1. EDT and ITDG Analysis by Averaging the Values of Each Row of Source Positions*

Considering the averaged results of each row of source positions, the graphs in Figure 16 show that the EDT values are higher by placing the source on the stage (Figure 16a) than in the pit (Figure 16b). Furthermore, the sound energy received in the stalls is higher than that received in the balconies. In addition, moving the source on the stage far from the audience, the EDT increases in both stalls and balconies. This effect is not found when the source is in the pit, obtaining steady EDT values in the stalls and decreased values when the source moves in the second and third row.

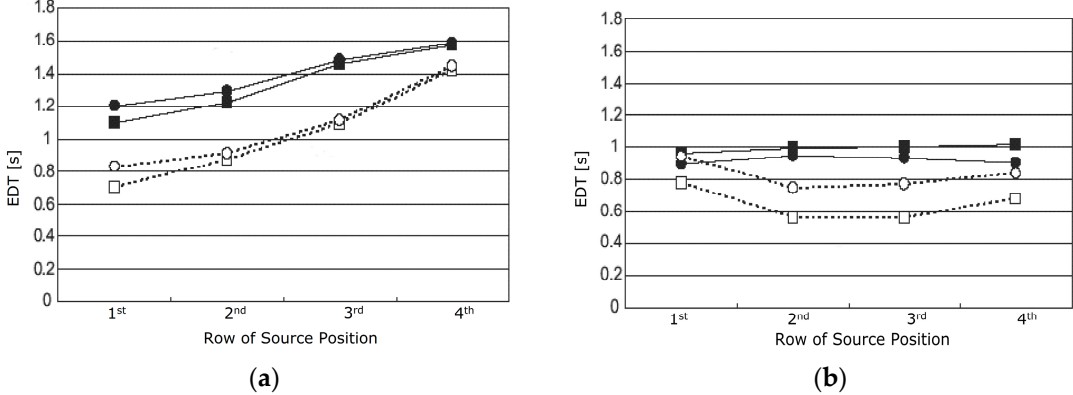

**(a)** **(b)**

**Figure 16.** Measurement of EDT during the second campaign of measurements in the Teatro Nuovo of Spoleto by placing the source on stage (**a**) and in the orchestra pit (**b**). Solid line represents the receiver placed in the stalls; dotted line represents the receiver in the balcony; circled mark represents the right channel; squared mark represents the left channel.

Another acoustic parameter subject to be considered for this type of study is the ITDG. Figure 17 shows the ITDG results with the source on the stage (Figure 17a) and inside the orchestra pit (Figure 17b).

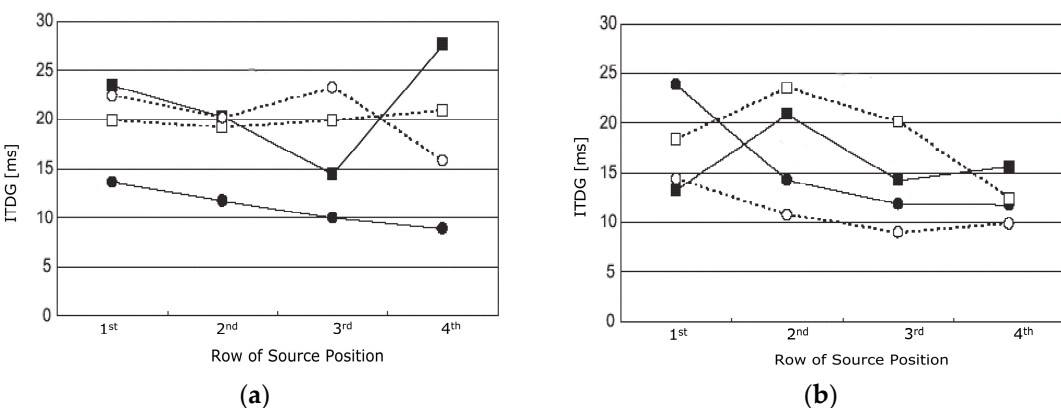

**(a)** **(b)**

**Figure 17.** Measurement of ITDG during the second campaign of measurements in the Teatro Nuovo of Spoletoby by placing the source on the stage (**a**) and in the orchestra pit (**b**). Solid line represents the receiver placed in the stalls; dotted line represents the receiver in the balcony; circled mark represents the right channel; squared mark represents the left channel.

Considering the average results of each row of source positions, Figure 17 shows that the reflections arrive approximately at the same time on the right and left channel by having the source on the stage and the receiver in the balcony. When the source is in the pit the time delay of arrival of the first reflection between the right and left channel is evident.

Moreover, Figure 17a shows that the sound reflections are shorter for the right channel placed in the stalls than the left channel. This result is due to the position of the dummy head in the right side of the main hall, located in a position close to the vertical walls (refer to Figure 15b).

In order to assess the directionality of reflections, it is convenient to broaden this study by analysing the results given by each source position of the first rows, regarding both the stage and orchestra pit.

The selection of the first rows of source position has been made based on their importance. They represent respectively the area of the stage where the singers act during the arias (in order to be closer to the audience), and the area of the pit related to string players (mainly soloists performing the principal melodies).

### 6.2. EDT and ITDG Analysis at Each Sound Source Position along the First Row on the Stage and in the Orchestra Pit

The selected parameters (i.e., EDT and ITDG) are influenced by the position of the receiver relative to the source. As such, Figure 18 shows the ITDG results obtained by placing the source on the stage (Figure 18a) and in the orchestra pit (Figure 18b).

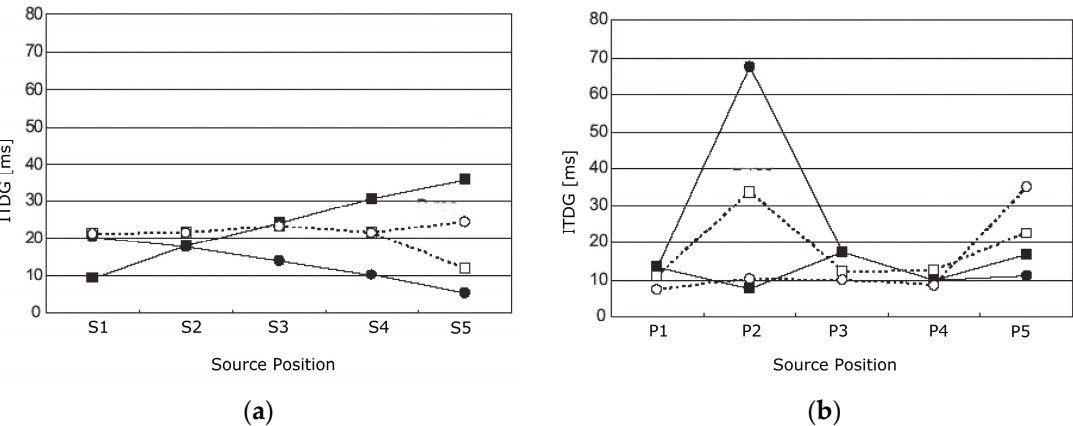

(a)                                                              (b)

**Figure 18.** Measurement of ITDG by placing the source along the first row of the stage (**a**) and along the first row of the orchestra pit (**b**). Solid line represents the receiver placed in the stalls; dotted line represents the receiver in the balcony; circled mark represents the right channel; squared mark represents the left channel.

Figure 18a shows that when the source is placed in position S1 on the stage, the receiver in the stalls registers a short ITDG for the left channel. At this position, the difference with the right channel is 10 ms. When the source is placed in position S5 the receiver in the stalls registers a short ITDG for the right channel, but the difference between the right and left channel is 28 ms. Furthermore, the results obtained in the stalls when the source is at the lateral position on the stage (i.e., S1 and S5) are lower than that measured when the source is in the center of the stage (i.e., S2, S3, and S4). This means that the vertical walls of the proscenium arch contribute to the first sound reflections.

When the receiver is in the balcony, the results do not show any difference between the left and right channel, floating at a constant value of 20 ms, except when the source is in position S5, closer to the balcony where the receiver stands.

Overall, the ITDG in the balcony results in a value greater than approximately 10 ms at every source position than that registered in the stalls. This could be attributed to the reflections given by other construction elements (e.g., ceiling, horizontal beam of the proscenium arch) that the receiver in the stalls gets more than standing in the balcony, where the sound comes mainly frontally [37].

Conversely, Figure 18b shows that the results registered in the stalls and the balcony are very similar to each other, with an ITDG of 33 and 68 ms relative to position P2, respectively revealed at the left channel of the balcony and at the right channel of the stalls.

A further analysis has been undertaken with the EDT parameter, as shown in Figure 19.

In Figure 19a, the EDT values in the stalls are higher than 0.5 s when the source is placed on the stage, which is found in the balcony with a difference of 0.2 s between the right and left channel being higher for the right channel. Figure 19b shows that the EDT values in the stalls are approximately 0.2 higher than the balcony when the source is placed in the pit, especially for positions P1 and P2, while they have similar results at positions P4 and P5, with a channel inversion for the stalls being higher at the left channel.

The data being analysed through the parameters of EDT and ITDG are limited for understanding the contribution of the ceiling to the selected receiver point in the stalls and on the third order of balconies. As such, a geometrical analysis is herein proposed for the behaviour of the early reflections hitting the ceiling of the main hall, as reported in Section 6.3.

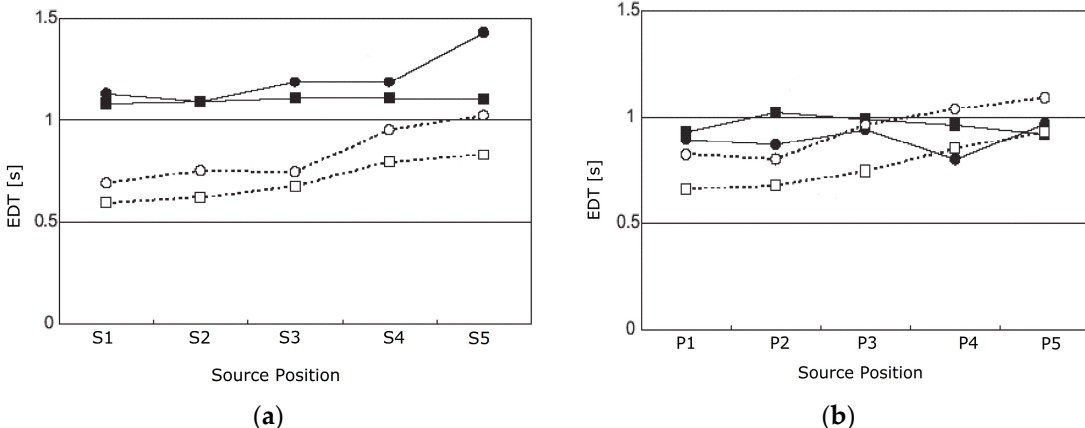

**Figure 19.** Measurement of EDT by placing the source along the first row of the stage (**a**) and along the first row of the orchestra pit (**b**). Solid line represents the receiver placed in the stalls; dotted line represents the receiver in the balcony; circled mark represents the right channel; squared mark represents the left channel.

*6.3. Geometrical Analysis of the Early Reflections Hitting the Ceiling at the Vertical-Longitudinal Section*

The image-source method (ISM) is often used as an alternative analysis technique in place of the more demanding digital model and its elaborated algorithms [38]. According to the specular reflection rules, a sound ray emitted by a source is reflected in such a way that the incident angle of the incoming ray is the same as the outgoing ray, since it is emitted by the source S′ as shown in Figure 20. The ISM assumes that the sound source is a point and the emitted rays have uniform energy in all the directions without any energy loss related to its time duration. Limitations of this method consist of ignoring the finished types of the boundary surface, which can contribute to scattering and diffusing the sound rays by having irregularities or different absorption coefficients on the flat plane [34]. The ISM is mostly applied for high frequencies, although it has demonstrated that this method grants very accurate results even at low frequencies [39].

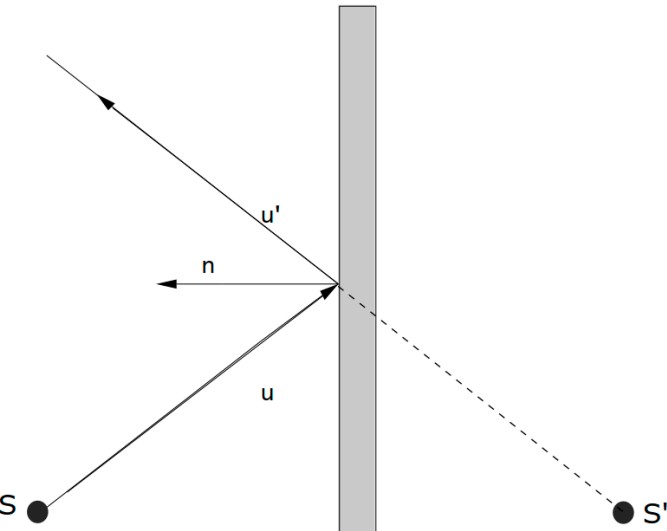

**Figure 20.** Representation of the image-source theory.

Other assumptions also include that the refraction, diffraction, and interference of sound waves against obstacles are to be considered neglected.

The simplicity of the image-source method is based on the visualization of the sound rays based on a geometrical approach only, as known also by the optics. Although this

method is not considered fully exhaustive for the aforementioned limitations, the ISM is a method still valid for understanding the acoustic behaviour of a space [40,41]. For the Teatro Nuovo of Spoleto, the ISM has been adopted since the ceiling is made of plastered plywood, which is considered as a hard surface finish.

Positions S3 and P3, located respectively on the stage and in the orchestra pit, are the two points of the emitted sound for this geometrical analysis, to be considered at its first and second orders of image-source. In order to study the early reflections given by the ceiling, the vertical section running along the longitudinal axis has been taken into consideration.

As illustrated in Figure 15, by applying the ISM from the S3 and P3 source positions and keeping the receivers in the same locations, it is possible to visualize that the early reflections (i.e., first and second orders) are not spread uniformly in the audience space.

In particular, when the source is in position S3 part of the ceiling which is close to the proscenium arch reflects the sound to the fourth order of balcony and to the gallery, while the other portions of the ceiling reflect the sound towards the second and third order of balconies. The second image-source reflections do not reach the receiver placed in the stalls, but rather direct the sound towards the stage and create an echoed effect [23].

When the source is in position P3 part of the ceiling which is close to the proscenium arch reflects the sound to the second order of balcony, while the other portions of the ceiling reflect the sound towards the back area of the stalls. In this configuration, the second image-source reflections bounce the sound towards the ceiling, except for a narrow stream of sound rays reaching the receiver placed in the stalls.

Figure 21 illustrates the early reflections inside the main hall with the existing ceiling.

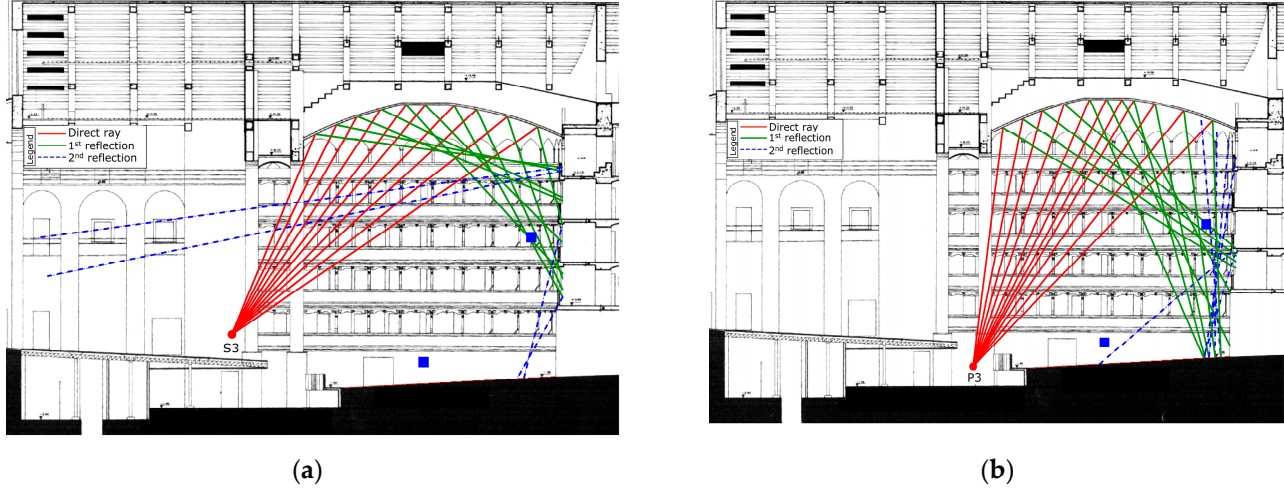

(**a**) (**b**)

**Figure 21.** Reflection configurations that resulted in the existing ceiling inside the Teatro Nuovo of Spoleto. Direct sound rays (continuous red line), the first order of reflections (continuous green line), and the second order of reflections (dashed blue line) are shown when the sound source is in positions S3 (**a**) and P3 (**b**).

In Figure 21, when the source is placed in positions S3 and P3, respectively on the stage and in the pit, the sound energy does not reach all the listening areas homogeneously. As such, the existing ceiling configuration provokes a concentration of the sound energy in specific locations, leaving other listening areas uncovered [11].

It is proposed by the authors to review the acoustics inside the Teatro Nuovo of Spoleto through considering the original ceiling configuration, which was 1.3 m lower in the central point than the existing one, with a curvature designed by keeping the same features at the perimeter [42]. Moreover, when the ISM is applied, the sound rays are more uniformly distributed, either at the first or second order of reflections [42].

Figure 22 illustrates the reflections inside the main hall by analysing the original ceiling existing before the refurbished works of the 20th century [11].

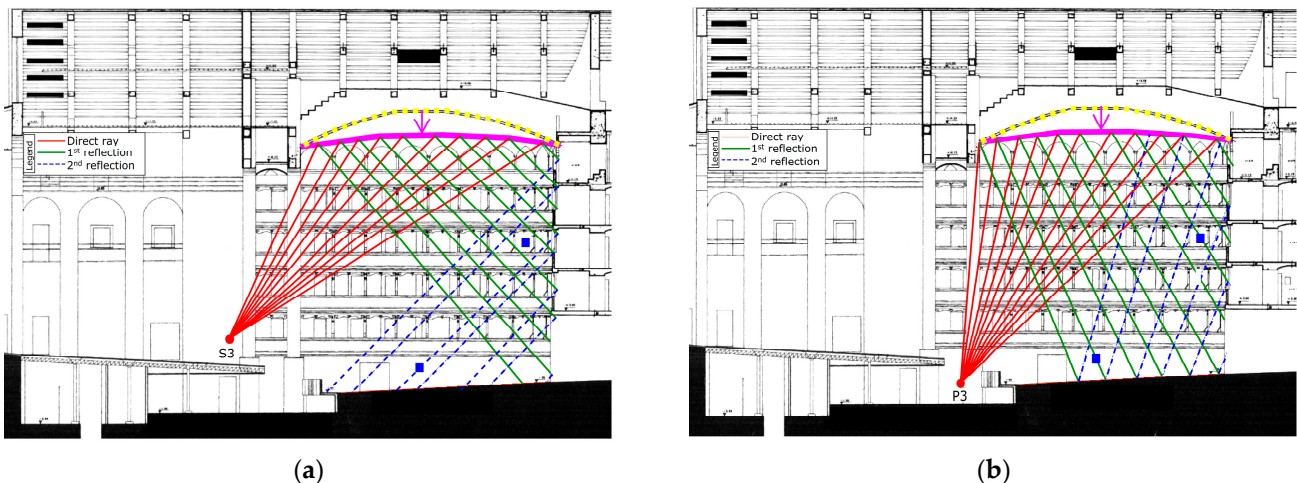

<center>(<b>a</b>)                                                                            (<b>b</b>)</center>

**Figure 22.** Reflection configurations that resulted in the original ceiling inside the Teatro Nuovo of Spoleto. Direct sound rays (continuous red line), the first order of reflections (continuous green line), and the second order of reflections (dashed blue line) are shown when the sound source is in positions S3 (**a**) and P3 (**b**). Dashed yellow line illustrates the existing ceiling height; violet continuous line indicates the original ceiling height.

## 7. Discussion and Conclusions

This paper presented the acoustical assessment of the Teatro Nuovo of Spoleto and Teatro Alighieri of Ravenna, two performance arts spaces of comparable shape and volume size. Although both theatres have a similar historical background, the decisions taken by the committees of the Teatro Nuovo to raise the ceiling by increasing the curvature of 1.3 m at the center, provoked acoustic alterations that caused the sound energy to be more concentrated in specific listening areas [11].

In the Teatro Alighieri, other matters that occurred regarding acoustic alterations, mainly focused on the resonance cavity. Fortunately, the decision of eliminating the cavity has been stopped in advance by researchers in acoustics of the University of Bologna, who proved experimentally the worsening acoustic conditions if the resonance cavity was filled with compressed polyester mats.

The damage and the preservation of the acoustic features, relative to the Teatro Nuovo and Teatro Alighieri, respectively can be mainly assigned to how the historical heritage would be preserved for future generations based on the architectonic values attributed by experts [43,44].

Regarding the Teatro Nuovo, further studies have been applied by placing the sound source in 20 positions on both the stage and orchestra pit. The results have been given in terms of EDT and ITDG. A final comparison between the existing and the original ceiling configuration, in relation to the sound energy distribution using the image-source method, has also been given. Future studies would be focused on the acoustics improvement across the listening areas of the theatre, and would be investigated more accurately using digital simulations and/or practical experimentations.

**Author Contributions:** Conceptualization, L.T.; methodology, L.T.; software, L.T.; validation, A.B.; formal analysis, L.T. and A.B.; investigation, A.B.; resources, L.T.; data curation, A.B.; writing—original draft preparation, A.B.; visualization, A.B. All authors have read and agreed to the published version of the manuscript.

**Funding:** This work was carried on within the project "SIPARIO-Il Suono: arte Intangibile delle Performing Arts–Ricerca su teatri italiani per l'Opera POR-FESR 2014-20", n. PG/2018/632038, funded by the Regione Emilia Romagna under EU Commission.

**Institutional Review Board Statement:** Not applicable.

**Informed Consent Statement:** Not applicable.

**Data Availability Statement:** Not applicable.

**Acknowledgments:** The authors wish to thank Alessandro Cocchi, Ryota Shimokura, Luca Lanciotti, Roberto Fiorilli, for their help during the measurements and the data processing.

**Conflicts of Interest:** The authors declare no conflict of interest.

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
