# Peer review of "Evaluation of Acoustic Features after Refurbishment Works Inside Two Historical Opera Theatres Located in Italy"

_acoustics, doi:10.3390/acoustics3020022_

Round 1

Reviewer 1 Report

Dear Authors,

The topic of the article - acoustic analysis of two theatres - is very interesting per se and it would be really nice and useful for the archeoacoustics and room acoustics communities to see this work published.

However, it is my opinion that the article in its current state needs much improvement to be considered for publication.

I think the research is conducted potentially correctly. Historical introduction is given; measurements are performed; analysis is performed on them; investigation about acoustics-related modifications is presented (the latter just for one theatre though).

However, each step needs thorough improvements from its current state: introduction must reference to the current state of the art in the field; measurements need to be described better; analysis must be improved since it is very basic for an original scientific publications aiming at adding something to the understanding of the acoustic field in these two theatres; section on acoustics-related modifications must be supported in a stronger way by scientific proofs, adding sound scientific grounds to all statements you make.

English language is sometime difficult to read and sometimes colloquial, more suiting a popular scientific publication rather than a peer-reviewed article for experts in the same field. I think your English is generally grammatically correct. Most improvement to the English language go together with improvements aiming at strengthening the scientific grounds of the article.

You use the word “commitment” a few times, where perhaps you would like to use “commette” (e.g. line 17).

Section 1

The introduction lacks any literature review, there are no references to state of the art contextualising the present work in respect to literature within archeoacoustics. Reference 5 is about one of the two studied theatres but there is no mention how your work refers to that reference . Objects, methods, analyses used in this article should be placed in reference to archeoacoustics and room-acoustics tradition, practices, state of knowledge - given also that these research fields are very much alive. Your most recent reference is from 2008.

Section 2

This section is about historical background of the two theatres. The language is colloquial at times (lines 61-72), English language should be improved (lines 65-69). Little if any reference to acoustics in section 2.1, ie how choices made during history have influence on acoustics - although later on in the article (see eg line 288-290) you make historical-acoustics statements.

Numbering at line 73 is wrong, not 2.1 but 2.2. Words such “similar story” or “young architects” (line 74, 77) are colloquial. Language in line 81-83 should be improved.

The statement about Teatro Alighieri being one of the few in the in the world preserving the resonance cavity (line 90-91) should be referenced and contextualised. “Proper” line 91: what is the definition of proper wooden hull?

“Construction been realized” line 92: improve language.

Statements line 93-96 need references, context.

Section 3

In general it is ok. Sometimes it is confusing because also Section 2 has information about architectural features. A better subdivision of text and presented concepts should be achieved. Underline relevant relationships between architecture and acoustics (see also line 288-290).

Line 124: by looking at figure 3 it is not possible to see a slight inclination of the floor of the stalls, it is a top 2D view, not a section. Clarify.

Line 138: by looking at figure 4 it is not possible to notice a moveable floor and mechanical system, it is a top 2D view, not a section. Clarify.

Line 143-144: perhaps those historical details should belong to section 2.2? Reconsider subdivision of text.

Table 1: “Excessively arched”, why excessively? Clarify and quantify.

The statements in line 153-154 and all second bullet line 162-167 must be clarified with scientific references to other articles/work/measurements/calculations performed either by the authors or by other researchers. You provide no scientific support to what you write. You write generic expressions such as “some” acoustics experiments, line 163, that should be improved language-wise.

It would be nice to have a better description of materials from their acoustical point of view.

Section 4

Add a clarification to what it is the scope of these measurements. What do you want to tell?

Introduce the section giving grounds to what you are going to present. Did you use any standard procedure? If so reference to ISO or similar standard. There are international standard for measuring reverberation times on field.

What is the high of source and microphone when you measured? I.e. distance from the floor.

The figure together with their caption should be standing independently (i.e. the reader should not need to read the text to fully understand the figure): add explanation to captions of figure 5 and 6 to what grey squares, number in b) figures, blue dot, S1, S2, represent. Unclear.

Section 5

The analysis is very basic and should be extended, also given reference 5 that concerns acoustic in Spoleto’s theatre.

  • It seems that you did many measurements. Present variations of results in respect to changing spatial locations of receivers, draw conclusions on that, e.g. by showing means versus standard deviations. What is the acoustics perceived at the stage by performers? By musicians? At the balconies? At the stalls? Those are interesting questions to be answered, I believe.
  • When you discuss differences between the two theatres you should reference those to JND, otherwise they may be empty statements. In this way, you quantify and objectify your analysis, rather than leaving it on qualitative and/or subjective levels.
  • You should distinguish language-wise between objective conclusions and subjective impressions (“feeling”, line 206). That may be done using JND. Or clearly refer to your educated impressions as authors and acousticians. 
  • Add vertical and horizontal lines to x and y axis to figures for readability.
  • Line 5.2: revise language.
  • You have two statements in line 210-212, saying that results on T20 are 1) due to architects and committees’s decisions and 2) because theatres should suit music and speech. What is the scientific base to those two statements? Historical? Other articles? Reference.
  • Same for 212-215 when you refer to refurbishment. Add scientific reference sustaining what you write.
  • Figure 8: add reference where this figure comes from. Add legend for red square and blue triangle.
  • line 219: difference is not null (0), but rather it is small. Revise language.
  • C50, C80: do comparison in terms of JND.
    • Line 222-224: you are contradicting yourself, from my point of view. First, you write that results indicate difficulties for speech at low frequencies, then you state that that it is not a negative results. Clarify.
    • Line 225: motivate/reference statement about 3 dB.
    • Line 226: “moderately”, this is qualitative statement, make it objective with reference to JND. Same for line 230.
    • Line 227-228: make reference, what does “generally” mean? Improve scientific ground to statement.
  • You present both C50 and D50, which are closely related. You seem do draw different or complementing conclusions in section 5.3 and 5.4 based on these two parameters. Curves for C50 and D50 in figure 7 look also different in respect to the two theatres. Why so? Clarify, this could be interesting to explain.
  • In theatres lateral reflections and immersion of the listener in the sound field is an important characteristic of the sound field. Any comment to that?
  • How do the decay curves look like?
  • Acoustical maps in section 5.5.
    • There is no mention on how these acoustical maps are derived. Are they calculation or interpolation of your measurement results? I think the latter is true, but clarify.
    • If it is interpolation, what grid size did you use?
    • Which measurements?
    • Noise maps are performed at a given high (meters): what is it in your case?
    • The colours in the figures are not defined, what are they representing? Why difference colours between various maps?
    • Lines in maps are isometric lines, are they? Clarify.
    • Why are you representing results at only 500 Hz? Clarify or at least mention the other frequencies. Motivate choice.
    • You show and mention for first time IACC: add discussion.
    • There is no discussion about what these maps are telling us, add discussion, analyse results, relate them to history and/or present architecture.
    • What do these noise maps tell differently from figure 7? Discuss.

Section 6

In general you present results in figures 12 and 13 and it is not clear how you devised them. Is it 3D ray tracing calculation? Is it paper and pen calculations on a 2D-section drawing?

It seems that you are drawing general results by looking at a very simple analysis performed on just one single vertical section of the theatre of Spoleto. If it is so, then your statements are weak scientifically.

Besides that, you are just looking at rays emitted along the elevation plane of the source, which is a large simplification - what about lateral reflections? Or reflections out of the elevation plane. (By elevation plane I mean the plane perpendicular to the floor/ceiling’s plane of the theatre, cutting the source of sound rays in its middle).

What about effect todiffusion/scattering (versus specular reflection, I mean)? Your lines in figures 12 and 13 assume perfectly specular reflection but perhaps the acoustic properties of ceiling allows for more diffuse reflections, at least at certain frequencies. Just a bit ofdiffusion/scattering may drastically change your conclusions.

The assumption of describing sound waves with geometrical rays works at certain frequencies only. What is the validity of the rays you drew in figures 12 and 13?

What would an image source analysis tell? You might want to think about that.

Here some more specific comments.

Line 253: when did you discuss that? Add internal reference for readability.

Line 255: now suddenly you mention sound rays - explain what you are doing. Sound rays are a simplification of geometrical acoustics with limitations and potentialities. Explain.

Line 256: vortices in acoustical maps? Clarify what you mean.

Line 265-267: what is the scientific proof to that? You write “clear” but I do not think it is clear, given that you show results just for a single section where all but one line of balconies and stalls are excluded from. “Unbalance listening”? How can you say so? Scientific proof? What about lateral reflections?

Line 268-271: all these statements on what would happen by a lowering of the ceiling (lowering of how much?) must be proved with a more advanced calculation, taking into account all the surface of the ceiling (and not just a line along a section), and later reflections. What would happened if you added diffusion/scattering to your analysis? Adding diffusion/scattering to the ceiling may well being a possible solution to improve acoustics by changing the ceiling.

Figure 13: you lower the ceiling, as green arrow represents. How much is that in meters? Clarify and quantify.

Line 275: revise language of title.

Line 276-278: contradiction. First, you write that lowering the ceiling is a solution to improve acoustics. Then you write that this solution is not viable due to artistic reasons (paintings). Does this also imply that all geometrical acoustic analysis in figures 12 and 13 have become irrelevant due to practical feasibility reasons?

Line 279-284: all of a sudden you present other solutions (reflecting panels etc) that are viable. You present no analysis or no calculation supporting your statements. A calculation, for instance in a room acoustic software for image source / ray tracing, must be added for strengthening the scientific value of your statements.

There is no discussion on the theatre in Ravenna. Why?

Section 7

Line 286: as I wrote, the acoustical assessment is very limited and must be improved.

Line 289-290: how do you know that previously (when?) Spoleto theatre had good acoustic given good listening previously to ceiling alternations? Reference.

Line 291-293: statements on resonance box have no ground. As far as I understand, you make no reference to resonance box in your acoustic analysis in previous sections. In figure 7 one may see lower D50 and C80 values fo Ravenna, but are those due to resonance box? You mention, compressed polyester, what’s its role? Complete, clarify.

References

The list is very short given the amount of research in the field. The most recent reference for peer-review article is from 2008, much has been produced since then. Complete reference list and Introduction accordingly.

Best regards.

Author Response

Dear Reviewer, thank you for sending your notes. The article has been revised. Please find attached the response to your comments. 

Reviewer 2 Report

Figure 1 and Figure 2

Insert other views of the theatres if possible

The resonance box was filled with polystyrene….

This fact is very interesting, it would be the case that the authors would report some more information, where it is located, what size if other theatres have the resonance boxes.

Acoustic measurements in the stalls and balconies. It would be advisable to insert in fig. 7 separate the values measured in the stalls from those measured in the balconies.

What happens in the presence of the audience?

Fig. 12 and Fig 13 how was it drawn? If the position of the source changes, especially in the orchestra, what happens?

Implement references, are too few.

Author Response

Dear Reviewer 2, thank you for your notes. The article has been revised based on your comments. Please find attached the response to your indications.

REVIEW 2

Figure 1 and Figure 2

Insert other views of the theatres if possible

 Dear Reviewer, additional pictures have been inserted.

The resonance box was filled with polystyrene….

This fact is very interesting, it would be the case that the authors would report some more information, where it is located, what size if other theatres have the resonance boxes.

 Dear Reviewer, additional information and descriptions have been given.

Acoustic measurements in the stalls and balconies. It would be advisable to insert in fig. 7 separate the values measured in the stalls from those measured in the balconies.

Dear Reviewer, the figures have been split as per your advice and suggestion.

What happens in the presence of the audience?

Dear Reviewer, measurements and data analysis of both theatres have been assessed without any audience. It would be subject to future investigations.

Fig. 12 and Fig 13 how was it drawn? If the position of the source changes, especially in the orchestra, what happens?

Dear review, data analysis by changing the source position has been given

Implement references, are too few.

Dear Reviewer, the list of references has been broadened.

Round 2

Reviewer 1 Report

Dear Authors,

thanks for the many improvements to the article. Your efforts are very much appreciated and the quality is much higher now, in my opinion. The topic is interesting, the new ISM analysis is very nice and effective in its simplicity (ie despite not using numerical methods).

Here come my comments to your latest version. Please refer also to the article pdf with markup where I highlight in yellow language/style-related comments and in blue scientific-related comments.

Line 3: capital O for Opera, whereas in the text you use little o. Inconsistent. 

Line 17: revise language.

Line 19: acoutics without s perhaps.

Line 20: revise language, something missing, like "...using the image-source methods to investigate a change...".

Line 23: Introduction is still missing a review of state of the art in the field, presentation of previous works related to these two theatres, identification of the research gap in the literature and explanation how your article fills this gap. There are 44 references now, but you discuss very little about them. What about ref 34-42 in line 42? Do they all refer to Teatro Nuovo of Spoleto? Or to study of different ceiling configurations? Probably not. Introduce them more precisely.

Best regards

Line 38: as far as I know, effort is a noun not a verb. Revise.

Line 47: "too" instead of very, perhaps.

Line 50: improve language.

Line 56: improve language.

Line 60: correct language.

Line 61: you often express numbers with digits and not in words. It is a style matter

Lines 59-65: is this paragraph on statues relevant?

Lines 69-73: revise language, long sentence.

Lines 73-74: language is heavy. "Additionally, matchinery to facilitate the operational activities of the stage was introduced". is in my opinion an easier-to-read variant.

Lines 74-77: reference to that?

Lines 84: verb tense not correct, you jump between tenses.

Lines 90: 1st World War.

Line 91: verb tense

Line 92: "widening of the stage" easier to read, I think

Lines 92-93-94: revise verb tenses.

Line 102: "and" is unnecessary and confusing, check language

Lines 108-109: simplify language

Line 127: why "even if"? What do you want to say?

Line 136: either the floor is inclined or not, not "should". "...where the floor of the stalls is slightly inclined". Language could be made easier to read.

Line 147-152: why does the separation of the first three rows of seats indicate that the pit floor can be lifted? Clarify.

Lines 152: "temporary"? in what sense. check language.

Line 153: it would be nice to understand why you focus so much on the comparison between the two theatres. I am not saying it is wrong, but what is the scientific goal of this comparison? What is it giving to the reader? What is it revealing?

Line 161: revise language more formal. "it is the case to mention that..." are empty words that should be avoided. If it the case to mention, just mention it. Start the sentence with "The restoration works...".

Lines 162-164: could be made lighter to read. E.g.: "This is explained in the following sections".

Line 170: this occurs a lot. You are inconsistent how you present numbers with units. The unit shall be separated from the number "13 m". This is recurring. I think only when you mention frequencies you write correctly, e.g. "50 Hz".

Line 171: language. "..., a space so deep that the reflections...".

Line 176: remove "the", I believe, unless the researchers from Bologna are just one group of people world-famous.

Line 179: check spelling.

Line 180-182: revise language and punctuation.

Line 186-196: why do you want to report the measurement equipment used by other people in another work? If one is interested they will check the reference. Am I missing something?

Line 197: maybe IRs since it is plural. Not sure.

Line 199 - 200: check verb tenses.

Line 205: I would use "two" not 2.

FIgure 5: are you reporting the plots from other researchers? Why? In the caption, give reference if so, but if they come from another article I do not see why you should report them here. Am I missing anything?

Line 213-214: clarify what is the aim of these measurements. 

Line 215-216: simplify language, e.g. "The measurements considered the thermo-hygrometric...".

Line 218 and 226: use same word.

Line 222: simplify language

Line 229: you already defined IR earlier.

Line 234: this is recurring. You are not consistent how you refer to figures. They have captions like "Figure X". Sometimes you write in the text Figure X or Fig X or Fig. X, same for plurals. Make it consistent.

Line 238: no blue dots or pentagons. Check spelling pentagons.

Figure 7: what are the numbers along the rows? You explain only the cardinal numbers 1st 2nd etc...

Line 245: theatre of Spoleto, not just Spoleto.

Line 246: "only"? what about the grey squares on the left-hand side?

Line 257: "on the stage"

FIgure 9: same as Figure 7.

Figure 10: it would be so much clearer if you plotted the JND, given all the discussion on JND the you do below. See e.g. Figure 10 in https://doi.org/10.3390/app11041586

Line 276: on what basis do you say that the difference is negligible? Refer to JND.

Line 279: often seats in theatres have built-in cushions not to change the absorption in the space if audience is missing, as you probably know. What you are writing is either wrong or incomplete (ie the seats are hard). Or, am i missing something? 

I guess also it is not "the increase of sound" but rather "the increase in reverberation time".

Lines 284 and FIgure 11: you write octave bands, refer to Figure 11 (line 286) but FIgure 11 has no octave bands. Clarify.

Line 287: "for reverberation time".

Line 287: can you prove that statement on committees?

Line 290: language. Of course that happens during some of the ten decades of 20th century, which ones?

Line 292: what about absorption surface areas? How do they come in the picture when extending the orchestra pit? Clarify.

Line 299: language is wrong.

Line 303: space, "3 dB".

Line 304: language.

Line 309: there is some confusion, I think, perhaps just linguistic. The JND for C80 is, I think, 1 dB. You say it is 1.5 dB, do you?

Line 310: I think you want to say that the difference between C80 at 4 kHz and the upper range limit of 2 dB is larger than the JND. RIght? Perhaps improve language?

Line 315: what is the JND for D50? Refine language, it is confusing. Are you saying that the difference between optimum values 0.5 and D50 is lower than JND for D50?

Line 333: "If"?! There is no if, the maps show what you describe.

Line 336: MATLAB.

Line 338. Coordinates "0.0", I guess (0;0) is a more common way of writing.

Line 342: reference, it is interesting.

Line 345 and Figure 12-13-14: you need to define a color scale, it is too imprecise to write e.g. "0 s (grey-pink".

Line 356: simplify language.

Line 358: "The IACC is defined in the literature [32] as....". Revise.

Line 360: consistency: earlier you do not refer to (a & b), now yes.

Line 361: verb tense.

Line 363: "owed" is wrong word choice, perhaps you want to say "due". owed refers to be in dept to something.

Line 387: be precise. either you remove because you assume the reader knows, or be precise specifying the decay of the first 10 dB of what.

Line 385 and 394: define ITDG before not after.

Line 408: check.

Line 412: refine language.

Line 426: refine language.

Line 427: use of "owed" is wrong.

Line 430: grammar. not ing-form.

Line 432-435: verb tense, simplify long sentence.

Line 438: what do you mean by time-dependant? EDT is frequency dependant, not time-dependant.

Line 498: move here limitation/comment on high frequencies in Line 509.

Line 505: "due to"

Line 507: "since"? would you have used another method if the ceiling was made of concrete?

Line 515: limitations due to that choice? Why that choice?

Line 555-556: you are hinting that the theatre of Ravenna does not have this problem because the ceiling was not risen. Or? If yes, you need to prove it. Or explain how you have proven it.

Line 563-564: theatres not cities.

Author Response

Dear Reviewer, thank you again for your observations that make this article more accurate. 

Your time has been very appreciated
